# Bioactive Ent-Kaurane Diterpenes Oridonin and Irudonin Prevent Cancer Cells Migration by Interacting with the Actin Cytoskeleton Controller Ezrin

**DOI:** 10.3390/ijms21197186

**Published:** 2020-09-29

**Authors:** Valentina Pagliara, Giuliana Donadio, Nunziatina De Tommasi, Giuseppina Amodio, Paolo Remondelli, Ornella Moltedo, Fabrizio Dal Piaz

**Affiliations:** 1Department of Medicine, Surgery and Dentistry “Scuola Medica Salernitana”, University of Salerno, Via Salvador Allende, 84081 Baronissi, Italy; vpagliara@unisa.it (V.P.); gamodio@unisa.it (G.A.); premondelli@unisa.it (P.R.); 2Department of Pharmacy, University of Salerno, Via Giovanni Paolo II, 84084 Fisciano, Italy; gdonadio@unisa.it (G.D.); detommasi@unisa.it (N.D.T.)

**Keywords:** ezrin, cell migration, cell Invasion, matrix metalloproteinase (MMP)-2, 9 activity, target identification, ent-kaurane diterpenes, proteomics, cancer metastasis

## Abstract

The ent-kaurane diterpene oridonin was reported to inhibit cell migration and invasion in several experimental models. However, the process by which this molecule exerts its anti-metastatic action has not been yet elucidated. In this article, we have investigated the anti-metastatic activity of Oridonin and of one homolog, Irudonin, with the aim to shed light on the molecular mechanisms underlying the biological activity of these ent-kaurane diterpenes. Cell-based experiments revealed that both compounds are able to affect differentiation and cytoskeleton organization in mouse differentiating myoblasts, but also to impair migration, invasion and colony formation ability of two different metastatic cell lines. Using a compound-centric proteomic approach, we identified some potential targets of the two bioactive compounds among cytoskeletal proteins. Among them, Ezrin, a protein involved in the actin cytoskeleton organization, was further investigated. Our results confirmed the pivotal role of Ezrin in regulating cell migration and invasion, and indicate this protein as a potential target for new anti-cancer therapeutic approaches. The interesting activity profile, the good selectivity towards cancer cells, and the lower toxicity with respect to Oridonin, all suggest that Irudonin is a very promising anti-metastatic agent.

## 1. Introduction

Several plant-derived bioactive compounds have anticancer properties [1,2]. Among them, ent-kaurane diterpenoids have certainly attracted particular attention among investigators because of their biological activity and safety profile [3]. Those compounds belong to a subclass of tetracyclic diterpenes containing a kaurane scaffold, with configurational inversion at all chiral centres, known as ent-kaurane diterpenes [3]. Oridonin (Ori) is a widely investigated compound derived from *Isondon rubescens*, a Chinese herb retaining various pharmacological features such as anti-inflammatory, anti-bacterial, and anticancer properties [4]. Therefore, Ori was one of the first diterpenoids undergoing clinical trial [3,4,5,6,7]. Several studies have reported that Ori prevents cell growth and invasion by inhibiting distinct signalling pathways in a wide range of cancer cells. Remarkably, Ori was shown to inhibit migration and invasion of U2OS osteosarcoma cells by decreasing the Matrix Metalloproteinases (MMP) expression [8]. Similarly, further evidences have shown that Ori was able to inhibit, in vitro, invasion of breast cancer cells either by reducing MMP-9 and MMP-2 expression, or by regulating the integrin β1/FAK pathway [9]. Again, in human breast cancer cells, Ori inhibits cell proliferation, migration, and invasion via a negative modulation of the Notch pathway, and reduces tumor mass in a nude mouse model [10].

Although the mechanism has yet to be elucidated, many evidences suggest that, for melanoma cells, Ori is a promising anti-metastastatic agent [11]. Indeed, recent papers have reported that Ori inhibits migration, invasion, and adhesion of A375 and B16-F10 melanoma cells. Remarkably, Ori could also negatively control the PI3K/Akt/GSK-3β signalling pathway and, in turn, TGF-beta-induced Epithelial to Mesenchymal Transition (EMT). This is a key event occurring during development, cell migration and cancer progression, which is associated with the reorganization of the actin cytoskeleton. Indeed, during EMT, cells reorganize their actin cytoskeleton and this event enables dynamic cell elongation and directional motility, thus increasing the migratory phenotype [12,13,14]. Other data indicate that Ori might accomplish its inhibitory effect by interfering with cytoskeleton dynamics and function. Indeed, the activity of myosin IIA, which is crucial for cell migration and invasion, is influenced in HeLa cells by the cell exposure to Ori, which impairs the interaction between myosin IIA and myosin phosphatase complex [15].

To investigate the molecular mechanisms underlying the inhibitory effect of ent-kauranic diterpenes on the invasion and migration processes in cancer cells, we tested the anti-invasive activity of Ori and of one its homologs, named Irudonin (Iru). Our findings evidenced that the two compounds were able to inhibit motility and the infiltrating ability of human melanoma A375 and gastric cancer MKN28 cells, and that such an ability was associated with the down-regulation of MMP-9 and MMP-2 activity. The results that we obtained also showed that both compounds, although with different efficacy, were able to interact with proteins involved in the actin cytoskeleton dynamics. In particular, we found that Iru interacts more efficiently than Ori with the protein Ezrin, which is implicated in multiple cellular processes involving actin cytoskeleton assembly, and as a consequence, cell motility and invasion.

## 2. Results

### 2.1. Exposure to Ori or Iru modifies Actin Cytoskeleton Organization and Inhibits Myotubes Formation in C2C12 Cells

With the purpose to investigate the effect of bioactive kaurane-diterpenes on cytoskeleton dynamics, we treated the C2C12 myoblast cell line, a reliable cell model to study cytoskeleton organization and function [16], with sub-toxic amounts of Ori, and of its structural homolog Iru (Figure 1A,B).

To that aim, we first evaluated the cytotoxic potential of both diterpenes on the C2C12 cells; therefore, we exposed for 24 h C2C12 myoblasts to increasing concentrations (10–60 μM) of Ori or of Iru, and, subsequently, we performed cell proliferation assay (Figure 1). Our results showed that both compounds induced a concentration-dependent reduction of the rate of cell proliferation when compared to control cells (Figure 1C). Specifically, Ori showed a 30 μM IC_50_ with respect to the 50 μM IC_50_ revealed by Iru, thus suggesting that Iru was better tolerated than Ori by C2C12 cells.

Next, we observed the consequences of cell exposure to Ori or Iru on the actin cytoskeleton formation occurring during C2C12 cells differentiation. This was monitored by evaluating myotubes formation and actin cytoskeleton organization by phase-contrast microscopy and by phalloidin staining of F-actin, respectively [17] (Figure 2). Phase-contrast microscopy (Figure 2A) revealed that control C2C12 cells developed myotubes of different size, following 48 h incubation in differentiation medium (DM). Cell exposure to Iru, and, to a lesser extent, Ori, inhibited myotube formation. Indeed, the mean diameter of myotubes, as well as the number of myotubes detected, was significantly reduced (*p* ≤ 0.001) in the Iru-treated C2C12 cells, with respect to control cells (Figure 2B). This result suggested that Iru, and, less efficiently, Ori, could interfere with the normal assembly of actin cytoskeleton in the early phase of C2C12 differentiation.

This hypothesis was confirmed by the observation that when the C2C12 cells were grown in DM supplemented with Iru, the expression level of the myogenin protein, an acknowledged marker of muscle differentiation, showed a significant decrease with respect to control cells (*p* ≤ 0.001) (Figure 2C). Ori treatment also induced a reduction of the myogenin levels, but with less effect than Iru. These data are consistent with the myotube differentiation experiment. Finally, fluorescence microscope analyses (Figure 2D) revealed a significant decrease of F-actin fluorescence intensity (*p* ≤ 0.001) and a strong reduction of multinucleated myotubes in the presence of Iru (Figure 2E) in comparison to untreated cells. Instead, cell treatment with Ori only slightly influenced the actin filaments and myotubes formation, indicating that the Iru is able to influence the assembly of actin cytoskeleton in the C2C12 cells more efficiently than Ori.

### 2.2. Ori and Iru Impair Migration Ability of Human Melanoma A375 and Human Gastric Adenocarcinoma MKN28 Cell Lines

Given the effect on the actin cytoskeleton organization revealed in C2C12 cells, we wondered whether the two ent-kauranic diterpenes could significantly affect morphology and migration of cancer cells. To this aim, we investigated the activity of Iru and Ori on human melanoma A375 and the gastric adenocarcinoma MKN28 cell lines, two cell lines showing remarkable metastatic property. Cell proliferation was assayed following the treatment of each cancer cell line with increasing concentrations of the two compounds (10–60 μM) for 24 h. The obtained results indicated that both the diterpenes inhibit proliferation of these metastatic cells in a concentration-dependent manner (Figure 3A,B), with Ori showing a higher toxicity (Appendix A). As expected, IC_50_ values of the two diterpenes measured for cancer cells (25 µM and 20 µM for Ori and 35 µM and 25 µM for Iru, towards A375 and MKN28, respectively) were significantly lower than those obtained for C2C12 cell lines. Based upon these results, and those obtained from apoptosis assays (Appendix A), the concentration of 10 µM and the exposure time of 24 h were chosen for the subsequent treatments for cellular and biochemical studies. To evaluate whether Ori and/or Iru could affect motility of the tumor cells, we performed wound closure assay [18] on un-treated, and Ori- or Iru-treated A375 and MKN28 cells. Results of wound closure assays showed that exposure to either Ori or Iru led to a significant reduction of migration speed in both cell types, as demonstrated by the larger wound area visualized at 24 h in the cultures of treated cell, compared to control ones (Figure 3C,E). Interestingly, wound areas of the Iru-treated cells were significantly larger than the ones produced by Ori (78% vs. 45% in A375 and 89% vs. 67% in MKN28). This differing effect produced by the two diterpenes was also observed in MKN28 cells after 6 h of treatment (Figure 3F). Conversely, the healing assay carried out on C2C12 cells showed that the treatment with the two compounds did not affect motility of these cells (Appendix A).

Taken together, these data indicated that Iru could efficiently impair motility of cancer cells, being less toxic than Ori, but more efficient in preventing cell migration.

### 2.3. Identification of Protein Targets by the Use of the Drug Affinity Responsive Target Stability (DARTS) Assay Coupled to Proteomic Analysis in A375 and MKN28

Next, we performed a proteomic-based study in order to identify Ori and Iru molecular targets that could explain the biological effect observed for these compounds in A375 and MKN28 cancer cells. For this purpose, we carried out DARTS assays. This is a compound-centered proteomic approach based on the evidence that the effective interaction of a protein with a ligand sensibly increases protein stability and reduces its proteolytic susceptibility. DARTS involves the incubation of small molecules with cells and/or cell lysates, followed by controlled proteolysis, SDS-PAGE, and LC-MS analysis, and allows studying the interactome of a bioactive compound without requiring its chemical modification and/or immobilization [19].

DARTS assays were therefore performed on in A375 and MKN28 cell lines, treated with amounts of Ori or Iru corresponding to the respective IC_50_ for 2 h. Using this approach, seven proteins emerged as putative ligands of the two compounds in both cell lines (Table 1). The presence, among them, of Hsp70 and Nucleolin- and its partner heterogeneous nuclear ribonucleoprotein-intrinsically validated our experimental results; these proteins were indeed previously identified as Ori-interacting proteins [19,20]. Intriguingly, three out of the other four identified proteins were involved in cytoskeleton organization and cell motility. Particularly, we found the Ezrin protein, a crucial factor for actin cytoskeleton organization at the plasma membrane boundary [21,22].

Given the key role played by Ezrin in the regulation of actin cytoskeleton organization, we thoroughly investigated the ability of the two diterpenes to interact with this protein in cancer cells. Firstly, we performed new DARTS experiments coupled to Western blotting analysis. These experiments revealed that Ezrin was significantly protected (2.8-fold, *p* < 0.001) from proteolysis in the A375 and MKN28 lysate derived from Iru-treated cells, compared to untreated cells (Figure 4A,B), confirming our previous finding. Similarly, Ori treatment produced a significant (2.2-fold, *p* < 0.001) protection of the Ezrin protein in MKN28, whereas in A375 cells the effect was less evident (1.6-fold, *p* < 0.05). Conversely, the incubation of C2C12 cells with Ori or Iru did not caused any detectable protection on Ezrin (Appendix A), suggesting that this protein may represent a target for the two compounds only in certain cancer cells.

In order to further ascertain the binding between Iru and Ori and Ezrin, we performed Cellular Thermal Shift Assay (CETSA), a cell-based test allowing us to carry out both qualitative and quantitative analyses of the direct interaction of a drug candidate to a target protein inside a cell [23]. CETSA relies on the principle that thermodynamic stabilization inferred to a protein caused by ligand binding can be used to monitor the formation of complexes within cells. The assay consists of the measurement of the shift in thermal stability of the target protein in the presence of the putative ligand, under different experimental conditions. The apparent melting temperature (Tm), observed in the presence or the absence of a bioactive compound, can be evaluated by measuring the persistence of soluble protein at different temperatures. Therefore, a typical output form CESTA experiment is the comparison between apparent melting measured in treated and control cells, and a potential thermal stabilization results in a shift of the melting curve to higher temperatures. Thus, we performed CETSA on A375 and MKN28 cell lines incubated with Ori and Iru for 2 h. After the treatments, Ezrin levels were evaluated by Western blotting analysis (Appendix A). Our result showed an increase in the Tm of Ezrin following the incubation with both compounds, passing from about 51 °C in the cells incubated with the vehicle to 53 °C and 54 °C in the cells treated with Ori or Iru, respectively (Figure 4C,D). In order to measure the EC_50_ of each compound in the two cell lines, we performed CETSA using different concentrations of the two diterpenes (from 5 to 35 µM). In these new experiments, we monitored the amount of soluble Ezrin following cell incubation at 53 °C, since at this temperature the efficacy of the treatments in stabilizing the protein was clearly detectable. Finally, we observed a dose-dependent stabilization of Ezrin by Iru and Ori in both A375 and MKN28 cell lines (Figure 4E,F). Again, Iru was revealed to be more efficient than Ori to stabilize Ezrin. Indeed, EC_50_ of 15 µM and 13 µM were estimated for Iru in A375 and MKN28 cells, whereas they were 19 µM and 21 µM for Ori.

### 2.4. Effects of Iru on Ezrin-induced AKT Phosphorylation, MMP-9 and MMP-2 Gelatinolytic Activity, Invasion Ability and Colony Formation in A375 and MKN28 Cells

Previous studies reported that Ezrin promotes migration, invasion and cancer progression *in vitro* and *in vivo*, by interacting with AKT thereby regulating the AKT pathway in breast cancer [14]. Thus, we investigated whether the ability of Ori and Iru to bind Ezrin in A375 and MKN28, may affect AKT signalling pathways. In order to prevent any anti-proliferative and/or pro-apoptotic effects from affecting the results, these experiments were conducted using sub-toxic quantities of the two substances. Based on our data (Figure 3A,B) and previously published results [11], we conducted the experiments by treating the cells with a 10 µm concentration of each compound. Resulting data showed that the treatment with the diterpenes was able to reduce the pAKT level compared to control cells (Figure 5A,B).

However, clear differences in the efficacy of the two treatments was observed, since in the presence of Iru a yield ~3-fold decrease of pAKT/AKT level compared with that observed in cells treated with Ori was detected.

Moreover, since Ezrin has been shown to modulate Matrix Metalloproteinases activity in cancer cells [9,14,24], we examined the effect of the two diterpenes on either MMP-9 or MMP-2 activity. To that aim we carried out gel zymography assay on cell-conditioned media from A375 and MKN28 cells, revealing (Figure 6A) in both cell lines that treatment with 10 μM of each compound was able to significantly reduce MMP-9 and MMP-2 gelatinolytic activity. Such a decrease was detected in both cell lines, being higher in MKN28 cell-conditioned media. Again, Iru produced a more evident effect on MMPs activity than Ori. Based on these data, we also explored the effect of either Ori or Iru on the infiltrating ability of A375 and MKN28 cells, by using Boyden chamber matrigel invasion assays (Figure 6B). Our experiments showed that both cell lines decreased their invasion ability when treated with 10 μM Ori or Iru for 24 h, compared to untreated cells; again, Iru treatment led to a significantly lower invasive ability (*p* < 0.001) than that observed in the presence of Ori (*p* < 0.01) (Figure 6C).

Cancer cells colony formation by Ezrin was reported [14,25]. Therefore, we assayed the reproductive rate of cancer cells following treatments with 10 μM Ori or Iru. To this purpose, we used the clonogenic assay [26], an in-vitro cell survival quantitative technique widely used to evaluate the capability of a single cancer cell to develop a large colony through clonal expansion. Our experiments showed that, upon exposure to Ori or Iru, the colony number of A375 and MKN28 cells was significantly reduced, compared to that observed in untreated cells. However, Iru treatment was able to completely abolish the growth rate and ability to form a colony of cancer cells (Figure 6D,E). Conclusively, these results strongly suggest that the binding of Iru to Ezrin may affect AKT phosphorylation, thus reducing the migration and invasion ability mediated by the modulation of MMP-9/2 activity in A375 and MKN28 cells.

## 3. Discussion

Several evidences suggested the potential of ent-kaurane diterpenes, and particularly of Oridonin, to inhibit cancer cell migration in both in-vitro [9] and in-vivo models [10]. These evidences prompt us to investigate the biological properties of two diterpenes, Oridonin, and its structural homolog, Irudonin, on different cell lines. In particular, we searched for molecular targets of either compound and we evaluated their potential to act as anti-metastatic agents on cancer cells.

Metastasis is a complex process responsible for the greatest number of cancer deaths. To generate metastatic lesions, malignant cancer cells utilize their intrinsic migratory ability to invade adjacent tissues from the primary site and ultimately metastasize. This complex process requires dramatic remodeling of the cell cytoskeleton. Indeed, reorganization of the actin cytoskeleton is the primary mechanism of cell motility and is essential for most types of cell migration. Aberrant regulation of the cytoskeleton plays an important role in the ability of cancer cells to survive chemotherapy, and mediate tumor cell migration, invasion and metastasis [13]. Since several studies showed that the inhibition of the actin reorganization decreases cell motility, elucidation of the molecular mechanism of this process is crucial for the finding of new therapeutic drugs. In this scenario, medical plants are an important source of medicines drugs. Indeed, plant-derived drugs became an indispensable source of anticancer agents, which are commonly used for treating tumors with the goal to improve therapeutic outcome of conventional pharmacological therapies, to alleviate their toxicity as well as to overcome multidrug resistance.

The main result of our work was the finding that Ezrin is a major target of both diterpenes in metastatic cells. This finding was obtained by using an innovative proteomic approach, based on in-cell interactomic studies and multiple biochemical assays. Ezrin is a part of the ERM (Ezrin, Radixin, Moesin) family of proteins and plays as linker between the plasma membrane and the actin cytoskeleton. As such, ERM proteins are placed at the centre of a regulatory network of many cellular processes, in both physiological and pathological conditions [22]. Indeed, the Ezrin protein is involved in multiple aspects of cell migration by acting both as cross-linker between the membrane, receptors and the actin cytoskeleton, and as regulator of signalling molecules that are implicated in cell adhesion, cell polarity and migration. Increasing evidence suggests that the regulation of cell signalling and the cytoskeleton by ERM proteins is crucial during cancer progression. More interestingly, a recent study indicates that Ezrin may serve as a metastasis-related oncogene that promoted the migration and invasion of cells during Epithelial-to-Mesenchymal Transition (EMT). Indeed, Li et al. indicated that Ezrin-regulated EMT via the AKT pathway [14]. In fact, down-regulation of Ezrin reduces AKT phosphorylation, while overexpression of Ezrin increased AKT activity. Taken together, these results demonstrated that AKT is required for Ezrin-mediated BC cells metastasis and angiogenesis. Additionally, the levels of MMP-9 and MMP-2 were decreased following Ezrin depletion [14,25].

In this context, it is worthwhile to mention that MMP-2, MMP-9, VEGF and VEGFR have been recently identified as the most common metastatic target proteins of *ent-kauranes* [3], although the mechanism of action through which they regulate these protein targets remains largely unknown. However, in many cancer cell lines, the MMP-2 and MMP-9 expression at both transcriptional and translational levels was regulated by PI3-K/AKT signalling pathway [27,28,29]. In light of these data, Ezrin represents a key regulator of tumor metastasis progression for its ability to regulate cell migration promoting EMT and to enhance the invasion capability through up-regulation MMP-2 and MMP-9 transcriptional and translational levels via the AKT signalling pathway.

In this scenario, the finding that Ori and, even more efficiently, Iru, are able to interact with Ezrin and to modulate Ezrin activity represents an important finding to take in account for new therapeutic approaches in the treatment or prevention of metastasis formation. Moreover, recent study indicated that Ezrin might serve as a metastasis-related oncogene that promotes the migration and invasion of cells during EMT via the Akt [14]. Notably, cancer cell treatment with Iru or Ori induced a marked reduction of Akt phosphorylation, and the same effect was previously observed when an Ezrin down-regulation occurred [14]. Additionally, the gelatinolytic activity mediated by MMP-9 and MMP-2 was clearly affected by the two diterpenes, and the level of these two proteases was shown to be decreased following Ezrin depletion [14].

Our results also allowed us to make an interesting comparison between the two investigated molecules. The two compounds differ only in the position of a hydroxyl group, and thus it is not surprising that they are able to interact with the same target proteins. However, some differences between the two diterpenes emerged. Firstly, Iru has a higher affinity than Ori towards Ezrin, and this determines its effectiveness in inhibiting the invasion capacity of tumor cells. On the other hand, Ori’s toxicity could be associated with its high affinity for proteins, such as Hsp70 and Nucleolin, which play a key role in tumor cell survival.

Together, these findings suggest that Iru could inhibit cell migration and invasion presumably through the inhibition of Ezrin and, consequently, through the impairment of the numerous pathways regulated by this protein. In agreement with these data, our results suggest that Ori, but especially Iru, was able to significantly reduce the phosphorylation of p-AKT in A375 and MKN28 cells. Moreover, the finding that in DARTS assays the exposure to Ori or Iru in C2C12 cell line did not cause any detectable protection of Ezrin indicated that this protein could represent a target for the two compounds only in cancer cells: an interesting suggestion to be verified in future work. Besides, it should be noted that the results of the proteomics experiments suggested the possible interaction of Ori and Iru with other proteins involved in the cell invasion process (i.e., Actin, Vinculin and Tubulin). These putative targets should be certainly further investigated, since if ent-kauranic diterpenes were able to modulate the activity of these proteins at the same time, they would have remarkable anti-metastatic potential. However, since these proteins interact directly or indirectly with Ezrin [30,31], it cannot be excluded that they emerged from the proteomic analysis only as secondary interactors of bioactive compounds.

In conclusion, our findings suggest that Iru might impair cell migration and invasion of metastatic cells throughout the inhibition of Ezrin and, consequently, of the several pathways regulated by this protein. The lower toxicity and the activity profile of Iru with respect to Ori makes Iru a suitable lead to design new anti-invasive molecules able to reduce the metastatic power of cancer cells. In fact, although the two diterpenes only differ because of the position of a single hydroxyl group, and therefore their activities were quite similar, Iru showed a higher affinity towards Ezrin in cancer cells and was more efficient in reducing cancer cell mobility and invasion than Ori. From this perspective, our study would provide valuable information about the design of novel anti-metastasis drugs.

## 4. Materials and Methods

### 4.1. Materials

Oridonin and Irudonin were extracted and purified (final purity >97%) from dry aerial parts of *Isodon rubescens*, kindly provided by Yee Poo International Company (Hong Kong), as reported elsewhere [32].

### 4.2. Cell Culture

Mouse C2C12 myoblasts, human melanoma cells A375 and human gastric adenocarcinoma cells MKN-28 were purchased from American Type Culture Collection (ATCC, Rockville, MD, USA). Cells were cultured in Dulbecco’s Modified Eagle’s Medium (DMEM, Gibco, Invitrogen, Grand Island, NY, USA) supplemented with 10% fetal bovine serum (FBS) and 1% penicillin/streptomycin (growth medium, GM) at 37 °C and 5% CO_2_. C2C12 myotube formation and cell differentiation were induced by using Differentiation Medium (DM) containing DMEM supplemented with 2% horse serum and 1% penicillin/streptomycin for 48 h. Drug treatment was performed for 24 h, following myotube induction by supplementing DM containing Ori or Iru for an additional 24 h. Drug exposure of cancer cells was performed for 24 h.

### 4.3. Cell Proliferation Assay

Cell proliferation rate was evaluated measuring mitochondrial metabolic activity through the use of PrestoBlueTM (PB) (Cat. N. A13262, Invitrogen), employed according to the manufacturer’s protocol [33]. Cells were plated onto 96-well plates (8000 cells/well), treated with increasing concentrations (10–60 μM) of Ori or Iru for 24 h and then incubated with PB reagent at 10% final concentration for 1 h. The absorbance was monitored at 570 nm with a reference wavelength set a 600 nm and using a microplate reader (Multiskan Go, Thermo Scientific, Waltham, MA, USA). Cell viability was expressed as a percentage relative to the un-treated cells cultured in DMEM with 0.1% DMSO and set to 100%. The dose compounds reducing cell number by 50% (IC_50_ values) were calculated using the respective dose–response curves according to the manufacturer’s protocol [20]. IC_50_ values were used for the subsequent proteomic experiments. Instead, the non-toxic concentration of 10 μM was used for the functional assays.

### 4.4. Apoptosis Assay

Annexin-V/Propidium Iodide (PI) assay was used to evaluate apoptosis rate during Ori or Iru treatment. Briefly, cells were seeded into a 6-well plate (2 × 10^5^ cells/well) and treated with 10 µM Ori or Iru, for 24 h. The control cells were grown in DMEM with 0.1% DMSO used as vehicle for Ori and Iru. After treatments, cells were washed twice with Cell Staining Buffer (Biolegend, San Diego, CA, USA) and were resuspended by gently vortexing in 100 µL Annexin-V Binding Buffer (Biolegend) containing 5 µL of Annexin-V/FITC and 10 µL of PI, for 15 min at RT. After incubation, 400 μL of Binding Buffer was added and samples were analysed using FACSVerse Flow Cytometer (BD Biosciences, San Jose, CA, USA). The antitumor agent, Paclitaxel (10 µM for 24 h) was used as positive control [34,35].

### 4.5. Drug Affinity Responsive Target Stability (DARTS) and Target Identification

The identification of putative targets of Ori and Iru were performed by Drug Affinity Responsive Target Stability (DARTS) experiments [19,20]. To identify target proteins A375 and MKN-28, living cells were first plated, and after their adhesion were incubated with compounds concentration corresponding to the respective IC_50_ for 2 h. After the treatments, cells were collected and whole protein extracts were prepared by lysing cells in 20 mM Tris-HCl (pH 7.5), 150 mM NaCl, 1 mM EDTA, 1 mM EGTA, 1% NP-40, 1% sodium deoxycholate, 2.5 mM sodium pyrophosphate, 1 mM β-glycerophosphate, 1 mM Na_3_VO_4_, 1 μg/ML leupeptin. Protein concentration was determined by the Bradford protein assay, using bovine serum albumin as standard. Identical amounts of proteins (50 μg) were subjected to a limited digestion with subtilisin (1:5000 *w*/*w*). The resulting partially hydrolyzed protein mixtures were separated by 10% SDS-PAGE. To get protein identification, the gels were divided into 10 pieces, and each underwent a trypsin in gel digestion procedure. NanoUPLC-hrMS/MS analyses of the resulting peptides mixtures were carried out on a Q-Exactive orbitrap mass spectrometer (Thermo Fisher Scientific), coupled with a nanoUltimate300 UHPLC system (Thermo Fisher Scientific). Peptides separation was performed on a capillary BEH C18 column (0.075 mm × 100 mm, 1.7 μm, Waters) using aqueous 0.1% formic acid (A) and CH3CN containing 0.1% formic acid (B) as mobile phases and a linear gradient from 5% to 50% of B in 90 minutes and a 300 nL·min^−1^ flow rate. Mass spectra were acquired over an *m/z* range from 400 to 1800. To achieve protein identification, MS and MS/MS data underwent Mascot software (v2.5, Matrix Science, Boston, MA, USA) analysis [36] using the non-redundant Data Bank UniprotKB/Swiss-Prot (Release 2020_03). Parameter sets were: trypsin cleavage; carbamidomethylation of cysteine as a fixed modification and methionine oxidation as a variable modification; a maximum of two missed cleavages; false discovery rate (FDR), calculated by searching the decoy database, 0.05. A comparison between the proteins found in the different samples allowed for discriminating those proteins remaining partially undigested in the compound-treated cells and largely hydrolyzed in the untreated control cells; those proteins were considered putative targets.

### 4.6. Cellular Thermal Shift Assay (CETSA)

The Cellular Thermal Shift Assay (CETSA) is well suited for identifying target engagement between ligands and their protein targets [37]. CETSA is based on the principle of thermodynamic stabilization inferred to the protein because of the ligand binding, which can be used for the estimation of interaction energies, as well other thermodynamic properties. The shift in thermal stability of the putative target protein is estimated by measuring the amount of remaining soluble protein at different temperatures for treated and control samples. A375 and MKN-28 cells were first plated and after their adhesion incubated with compounds concentration corresponding to their respective IC_50_ for 2 h. After the treatments, cells were collected and the samples were divided in 6 aliquots, each of them then subjected to a 5 min incubation at a specific temperature, in the range from 45 to 60 °C. Samples where then lysed and centrifuged, in order to separate the soluble proteins from the aggregated and precipitated ones. The amount of soluble protein target was evaluated by Western blotting analysis. To establish the half-maximal effective dose (EC_50_) of Ezrin/Iru or Ezrin/Ori complex, A375 and MKN-28 cells were first plated and after their adhesion were exposed to different concentrations of Ori and Iru (5, 10, 15, 20, 25, 30 and 35 μM) for 2 h. Following the incubation, cells were heated at 53 °C (Tm of complexed Ezrin) for 5 min. The amount of soluble protein target was evaluated by Western blotting analysis. For each compound, the density ratio Ezrin/GAPDH measured at the compound concentration producing the maximum stabilizing effect was set at 100%.

### 4.7. Phalloidin Staining and Phase-Contrast Analysis

C2C12 cells grown on glass coverslips were cultured in GM for 24 h. Differentiation of myoblasts into myotubes was induced by switching the medium from GM to DM. To perform immunofluorescence analysis, undifferentiated and differentiated cells were washed in phosphate-buffered saline (PBS), fixed in PBS-4% paraformaldehyde, and permeabilized with 0.1% Triton X-100 in PBS for 5 min [38]. Thereafter, cells were stained with a Phalloidin-Tetramethylrhodamine B isothiociyanate (1:5000) (Merk, Darmstadt, Germany) for 20 min and 4’,6-diamidino-2-phenylindole (DAPI) to visualize the nuclei. After washing, coverslips were mounted with Vecta-mount medium. Images were acquired with a fluorescence microscope (Nikon Eclipse Ti, Tokio, Japan) with a 40× objective. The microscope was equipped with a digital video camera (Q-Imaging Retiga 2000R; Tucson, AZ, USA) with digital image software (NIS-Elements AR Analysis 4; Nikon). Phase-contrast images were captured using a Leica DM IL LED inverted microscope (10× objective) (Meyer Instruments, Inc., Huston, TX, USA). The myotube diameter was quantified as follows: 10 fields were chosen randomly, and 10 myotubes were measured per field. The mean diameter per myotube was calculated as the means of the 3 measurements taken along the length of the myotube [39]. The fluorescence intensity and myotube diameters were measured using the free image-processing software ImageJ, version 1.47 (http://rsb.info.nih.gov/ij/) [40]. Quantitative analyses were performed selecting an entire field as region of interest (ROI), quantifying the mean pixel intensity. To reduce intrinsic variability, we repeated this measurement on at least thirty fields per experimental point. Then, the fluorescence intensity was relative to untreated cells cultured in DM and set to 100%.

### 4.8. Western Blotting Analysis

Whole protein extracts were prepared by lysing cells in 50 mM Tris–HCl pH 8.0, 150 mM NaCl, 0.5% sodium deoxycholate, 0.1% SDS, 1 mM EDTA, 1% Igepal, 1× protease inhibitor and phosphatase inhibitor cocktail. Protein concentration was determined by the Bradford protein assay using bovine serum albumin as standard; identical amounts of proteins (30 μg) were subjected to 10% SDS-PAGE. After electrophoresis, proteins were transferred to a nitrocellulose membrane and then incubated with the specific primary antibody. The following antibodies were used: rabbit polyclonal antibody raised against Myogenin (1:1000, Santa Cruz Biotechnology, Dallas, TX, USA), mouse monoclonal antibody raised against Ezrin (1:1000, sc-71082, Santa Cruz Biotechnology), rabbit polyclonal antibody raised against pAKT (Ser473) (1:1000, sc-7985-R, Santa Cruz Biotechnology), mouse monoclonal antibody raised against AKT (1:1000, sc-81434, Santa Cruz Biotechnology), mouse monoclonal antibody raised against GAPDH (1:1000, sc-32233, Santa Cruz Biotechnology).

After incubation with the appropriate anti-rabbit or anti-mouse (1:5000, Pierce, Thermo Fisher Scientific) peroxidase-linked secondary antibody, detection was achieved using Enhanced Chemiluminescence (ECL) kit (GE Healthcare, Chicago, IL, USA). Densitometry analysis was performed using the free image-processing software ImageJ, version 1.47 (http://rsb.info.nih.gov/ij/).

### 4.9. Wound Closure Assay

The rate of cell migration was assessed by using the cell wound closure assay [41,42]. Briefly, cells (2 × 10^5^ cells/well) were seeded into each well of a 6-well plate and incubated with complete medium at 37 °C and 5% CO_2_. After 24 h of incubation, the cells were scraped horizontally and vertically with a sterilized pipette tip to form a cross shaped wound and then subjected to the different drug treatments in medium supplemented with 0.5% FBS in order to overcome the generic proliferative effect of the serum. Quantitative analysis of the closure assay was performed by measuring the gap area within the wound using the free image-processing software ImageJ, version 1.47. To ensure the reproducibility of the wound, we scratched with the pipette tip and applied constant pressure to create a constant gap width, repeating each single experiment at least in triplicate. Moreover, to photograph the same field during the image acquisition, we scraped horizontally and vertically with a pipette tip to form a cross-shaped wound. Cross-shaped wounds were photographed on each well at 0, 6, 24 and 48 h by using an inverted microscope (10× objective) and then quantitative analysis of the closure assay was performed by normalizing the gap area with the time 0 for each group. Then, after normalization, the wound closure was expressed as percentage vs. untreated control cells.

### 4.10. Invasion Assay

Cell invasion assay was performed by BD Falcon BioCoat Matrigel invasion chambers (cat. n. 354480, BD Bioscience). The cell culture inserts were rehydrated and prepared as described in the manufacturer’s instructions [42]. Briefly, 2 × 10^4^ cells in 0.5 mL of DMEM with 0.5% FBS were seeded in the upper chamber, and 750 μL medium with 5% FBS was placed in the lower chamber. After 24 h, cells in the upper chamber were removed with the cotton swab, and the cells at the bottom of the filters were fixed and stained with a Diff-Quick kit (cat. n. B4132-1A, Becton-Dickinson). The inserts were allowed to air dry, and phase-contrast images were captured using a Leica DM IL LED inverted microscope (10× objective) (Meyer Instruments, Inc.). Cell invasion ability was determined by counting cells in five randomly selected fields per membrane using the free image-processing software ImageJ, version 1.47. Cell number was expressed as percentage vs. untreated control cells.

### 4.11. Gelatin Zymography

Gelatinolytic activity in the cell-conditioned medium was assayed by SDS-PAGE zymography, as described previously [43]. Samples were analyzed under nonreducing conditions without boiling, through a 10% SDS-polyacrylamide gel co-polymerized in the presence of gelatin (1 mg/mL, Merk). Electrophoresis was conducted at 35 mA for 90–120 min at 4 °C. After the run, the proteins in the gels were renatured in a 2.5% Triton X-100 solution for 1 h. The gels were then incubated with 50 mM Tris-HCl pH 7.5, 200 mM NaCl, 5 mM CaCl_2_, and 5 μM ZnCl_2_ at 37 °C for 48 h, which allows substrate degradation. Finally, the gels were fixed in 30% methanol, 10% acetic acid for 30 min, and then stained with 0.5% Coomassie Brillant Blue R-250. Proteolytic bands were visualized by destaining with 50% methanol and 5% acetic acid and the densitometric analysis was performed using the free image-processing software ImageJ version 1.47 (http://rsb.info.nih.gov/ij/).

### 4.12. Colony-Formation Assay

For colony-formation assays [26], cells were treated with 10 μM Ori or Iru. After 24 h, cells were re-plated 5 × 10^3^ on 6-well plates in triplicate. The medium was replaced, and after incubation for 7–10 days, when the colonies were visible to the eye, the culture was terminated by removing the medium and washing the cells twice with PBS. The colonies were fixed with 95% ethanol, then dried and stained with 0.1% crystal violet solution for 20 min, and then washed with PBS. Images were obtained and the number of colonies was counted with the free image-processing software ImageJ, version 1.47 (http://rsb.info.nih.gov/ij/). Each treatment was performed in triplicate. Cell number was expressed as percentage vs. untreated control cells.

### 4.13. Statistical Analysis

Statistical significance was determined by one-way analysis of Variance (ANOVA), followed by Bonferroni test. Each value represents the mean ± SD of at least three independent experiments performed in triplicate (* *p* < 0.05, § *p* < 0.01, # *p* < 0.001).

## Figures and Tables

**Figure 1 ijms-21-07186-f001:**
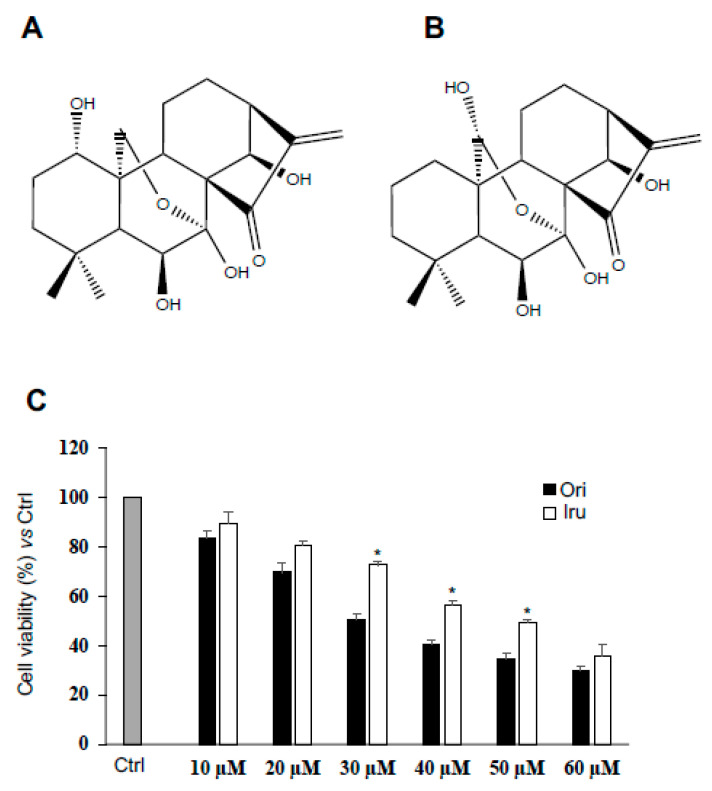
Effect of Ori and Iru exposure on C2C12 cell proliferation. (**A**) Chemical structure of Oridinin (Ori) ((1S,2S,5S,8R,9S,10S,11R,15S,18R)-9,10,15,18-tetrahydroxy-12,12-dimethyl-6-methylidene-17-oxapentacyclo (7.6.2.15,8.01,11.02,8) octadecan-7-one). (**B**) Chemical structure of Irudonin (1S,2S,5S,8R,9S,10S,11R,15S,18R)-9,10,17,18-tetrahydroxy-12,12-dimethyl-6-methylidene-17-oxapentacyclo (7.6.2.15,8.01,11.02,8) octadecan-7-one) (**C**) Results of cell proliferation assay on C2C12 cells. Cells were exposed to increasing concentration of Ori and Iru ranging from 10 to 60 µM as indicated for 24 h and then incubated with (PB) reagent at 10% final concentration for 1 h. Histograms represent percentage (mean ± SD) (*n* = 6) of the control cells, cultured in DMEM with 0.1% DMSO, set as 100%. Columns with (*) were statistical significantly different from Ori treated cells (* *p* < 0.05).

**Figure 2 ijms-21-07186-f002:**
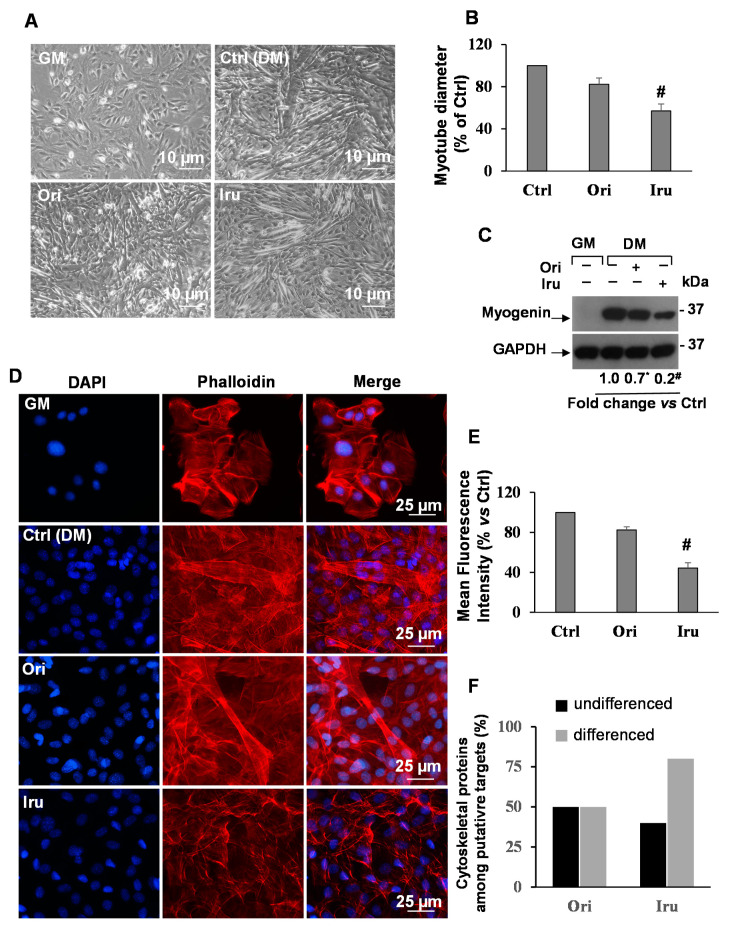
Effect of Ori and Iru exposure on myotube formation and actin cytoskeleton organization in C2C12 cells. (**A**) Phase-contrast micrographs of C2C12 cells cultured in GM, DM or exposed to 10 μM of Ori or Iru. All the treatments were performed for 24 h in presence of 0.1% DMSO, used as vehicle for Ori and Iru. Scale bar = 10 μm. (**B**) Quantitative measurements of mean diameter of myotubes. Histograms represent mean % ± SD (*n* = 6), with respect to the Ctrl cells, set as 100%. # indicates values statistically different from control (# *p* < 0.001). (**C**) Western blotting showing Myogenin protein expression levels. GAPDH was used as loading control for cell lysates. Fold change in Myogenin levels was calculated by first normalizing to GAPDH levels in individual samples and then relative to un-treated control (cells cultured DMEM with 0.1% DMSO, vehicle) set as 1. # and * indicate values significantly different from Ctrl (# *p* < 0.001; * *p* < 0.05). Statistical analysis of the results obtained in triplicate experiments are reported in the Appendix A. (**D**) Representative fluorescence images of C2C12 myoblast cells cultured in GM, DM or exposed to 10 μM of Ori or Iru for 24h. Cells were subjected to fluorescence analysis with TRITC-coupled phalloidin (red). Nuclei were stained with DAPI (blue); 0.1% DMSO was used as vehicle for Ori and Iru. Scale bar = 25 μm. (**E**) Quantification of fluorescence intensity. Results are presented as percentage (mean ± SD) (*n* = 6) of the control cells, cultured in DM with 0.1% DMSO (vehicle), set as 100%. # indicates values statistically different from control (# *p* < 0.001).

**Figure 3 ijms-21-07186-f003:**
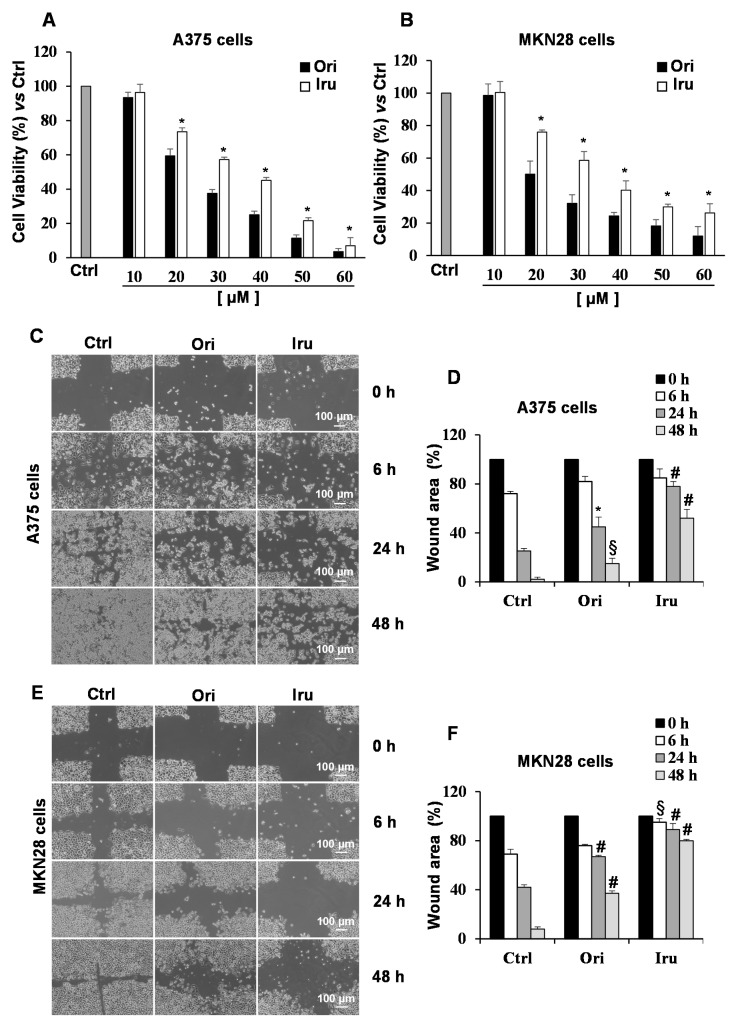
Effect of Ori and Iru exposure on A375 and MKN-28 cell proliferation and cell migration. (**A**) A375 and (**B**) MKN28 cell proliferation was quantitated by the cell viability reagent (PrestoBlue, PB) at 24 h. Results are presented as percentage (mean ± SD) (*n* = 6) of the control cells, cultured in DMEM with 0.1% DMSO, set as 100%. (*) indicates histograms statistically different from Ori treated cells (* *p* < 0.05). (**C**,**E**) Representative phase-contrast-microscope (10× objective) images of the wound healing assay on A375 and MKN28 cells. (**D**,**F**) Quantification of wound area was performed using the free image-processing software ImageJ, version 1.47. For each treatment, data show the wound area at the indicated time in comparison to that of the open wound at time 0, set as 100%. Results are presented as mean ± SD (*n* = 9). Columns with (*, §, #) were statistically different from untreated control cells (* *p* < 0.05, § *p* < 0.01, # *p* < 0.001).

**Figure 4 ijms-21-07186-f004:**
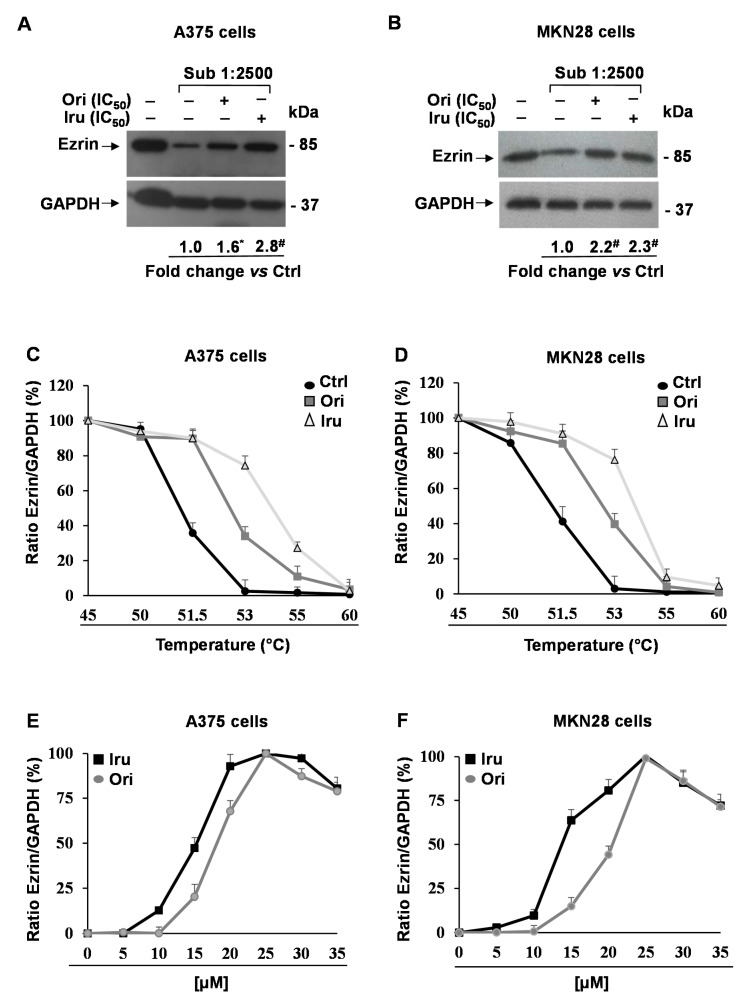
Identification of Ori and Iru protein targets in A375 and MKN28 cell. Western blotting showing Ezrin protein levels in (**A**) A375 and (**B**) MKN28 cells treated with IC_50_ values of Ori or Iru for 2 h. After treatments, cells were subjected to a limited digestion with subtilisin and partially hydrolyzed protein mixtures were used for Western blotting analysis of DARTS experiments. GAPDH was used as a control protein as it partially resists to subtilisin-catalyzed proteolysis. Fold change in Ezrin levels was calculated by first normalizing to GAPDH levels in individual samples and then relative to un-treated control set as 1. (*, #) indicate statistical significance toward control (* *p* < 0.05; # *p* < 0.001). Statistical analysis of the results obtained in triplicate experiments are reported in the Appendix A. CETSA melting curves of Ezrin in (**C**) A375 and (**D**) MKN28 cells treated with IC_50_ values of Ori or Iru for 2h and then subjected to 5 min incubation at the indicated temperature (45–60 °C). Ezrin level at 37 °C was set at 100%. (**E**) A375 and (**F**) MKN28 cells were treated with different amounts (range 5 μM–35 μM) of Ori or Iru for 2h and then subjected to a 5 min of incubation at 53 °C. Ezrin levels were evaluated by Western blotting analysis. Densitometry-based quantification of Western blotting signals was calculated by first normalizing to GAPDH levels in individual samples. Data were reported as Ezrin amount increment (%) in respect of untreated cells. Maximum Ezrin levels reached in each experiment was set at 100%.

**Figure 5 ijms-21-07186-f005:**
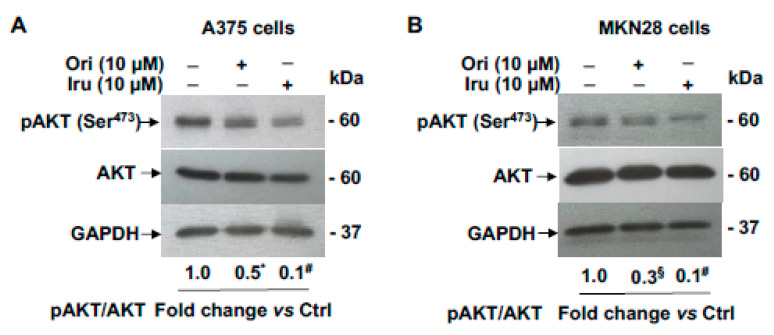
Effects of Ori and Iru exposure on AKT phosphorylation in A375 and MKN28 cells. Western blotting analysis of phospho-AKT expression levels in (**A**) A375 and (**B**) MKN28 cells treated with Ori and Iru for 24 h. The relative fold change vs. untreated cells, set as 1, of protein levels is shown under each lane. Values marked with *, § or # were statistically different from control (* *p* < 0.05; § *p* < 0.01, # *p* < 0.001). Statistical analysis of the results obtained in triplicate experiments are reported in the Appendix A.

**Figure 6 ijms-21-07186-f006:**
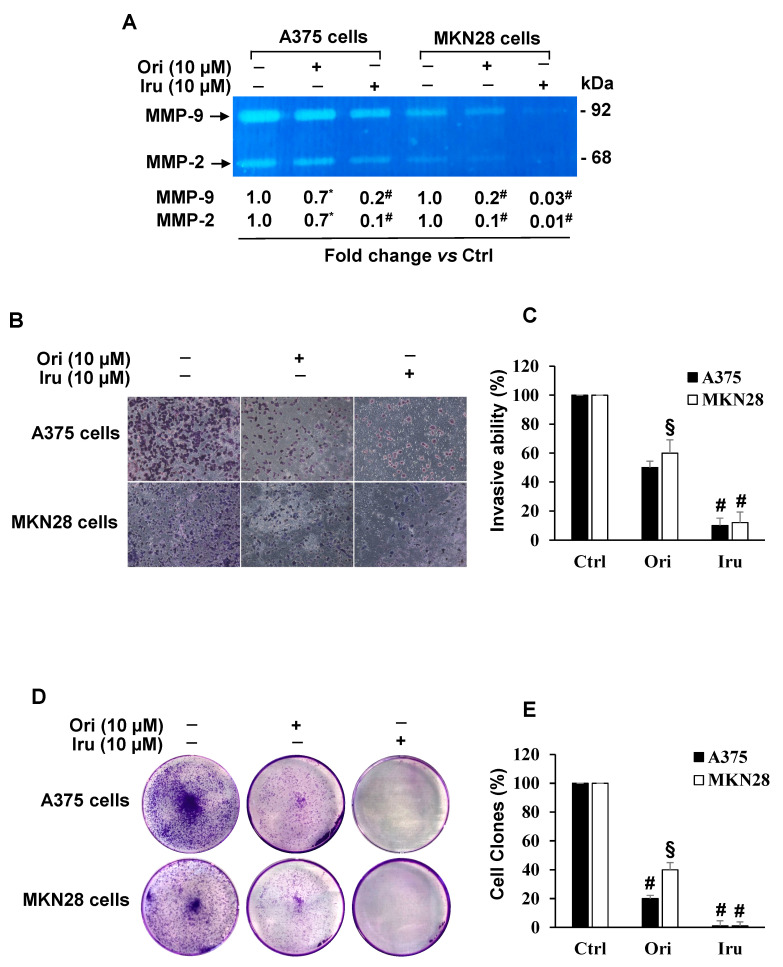
Effect of Ori and Iru exposure on A375 and MKN-28 cell invasion, MMP-9 and MMP-2 activity and colony formation. (**A**) Gelatin zymography of MMP-9 and MMP-2 in A375 and MKN-28 cells. Fold change in MMP-9 and MMP-2 activity was calculated by first normalizing with Ponceau-S staining of membranes (Control of protein loading) in individual samples and then relative to untreated control (cells cultured DMEM with 0.1% DMSO, vehicle) set as 1. The numbers with (*, #) were statistically different from control (* *p* < 0.05; # *p* < 0.001). Statistical analysis of the results obtained in triplicate experiments are reported in the Appendix A. (**B**) Representative phase-contrast-photomicrographs (10 × objective) of random fields of matrigel invasion assay in A375 and MKN28 cells. (**C**) The invasive capability was determined by cell counting in five fields, randomly selected, per membrane. Quantification was relative to untreated cells, cultured in DMEM with 0.1% DMSO, vehicle, set as 100%. Results are presented as mean ± SD (*n* = 9). Columns with (*, §, #) were statistically different from untreated control cells (* *p* < 0.05, § *p* < 0.01, # *p* < 0.001). (**D**) Representative images of clonogenic analysis in A375 and MKN28 cells. (**E**) The colony number quantification was relative to untreated cells, cultured in DMEM with 0.1% DMSO, vehicle, set as 100%. Results are presented as mean ± SD (*n* = 9). Columns with (§, #) were statistically different from untreated control cells (§ *p* < 0.01, # *p* < 0.001).

**Table 1 ijms-21-07186-t001:** Putative Ori and Iru targets identified in both the cancer cell lines by mass spectrometry-based Drug Affinity Responsive Target Stability (DARTS) experiments.

Protein (Swiss-Prot CODE)	MNK28		A375	
	ORI	IRU	ORI	IRU
Alpha actin (ACTN1_HUMAN)	X	X	X	X
Heat shock cognate 70 (HSP74_HUMAN)	X	X	X	X
Nucleolin (NCL_HUMAN)	X	X	X	X
Ezrin (EZRI_HUMAN)	X	X	X	X
Heterogeneous nuclear ribonucleoprotein L (HNRL_HUMAN)	X	X	X	X
Alpha tubulin (TBA1B_HUMAN)		X	X	X
Beta-enolase (ENOB_HUMAN)	X	X	X	X
Vinculin (VINC_HUMAN)		X		X

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
