# Peer review of "Bioactive Ent-Kaurane Diterpenes Oridonin and Irudonin Prevent Cancer Cells Migration by Interacting with the Actin Cytoskeleton Controller Ezrin"

_ijms, 2020, doi:10.3390/ijms21197186_

Round 1

Reviewer 1 Report

Review of Pagliara, et al., 2020. International Journal of Molecular Sciences

In this article, Pagliara and colleagues investigate the effects of two dieterpenes, Ori and Iru, on cancer cells. It has been noted previously that they have anti-metastatic effects. The authors show that these compounds inhibit proliferation, differentiation and invasion. Importantly, they identify Ezrin as a molecular target that may mediate their metastatic inhibition.

The manuscript is well-written (although the English should be checked at a few points) and the conclusions are justified based on the experimental results. The identification of Ezrin as a target of these compounds is an important finding. I am happy to recommend this manuscript for publication. I only have two major comment, and the rest are minor.

Major Comments

The authors should do an apoptotic assay to test that the concentrations of compound they are using are not causing apoptosis to any significant degree.

Scratch assays are known to cause cell damage in wound healing assays. I suggest the authors should perform a wound healing assay where an intermediate block is removed and the cells allowed to migrate into the space.

Minor Comments

Lines 124-125 – the authors say that Iru and Ori delay myotube formation. However, there is no evidence that this is delayed: this would require subsequent time point images that show myotube formation with a delayed effect. I suggest the authors provide such images of different time points, or otherwise change the conclusion to an inhibition of myoutube formation rather than a delay.

Lines 130-133 – the authors should say that Ori also had the same effect, and that the lesser effect with Ori (compared to Iru) in the Western blot is consistent with the myotube differentiation experiment.

Line 206 – a brief explanation of DARTS in the main text would be useful for non-specialists

Figures with blots – on all westerns, fold changes are written which is useful. But no errors/variation are included. It would be useful to know that such a change is significant across multiple experiments – this could be included as a supplementary bar graph with error bars, for example.

Discussion – the authors find a number of interesting candidates (table 1) but do not discuss them in their discussion section. A discussion of these would be very useful for the reader, as many of these candidates are known to be important in invasion (e.g. vinculin, tubulin, actin).

Author Response

The authors thank the reviewer for his suggestions. we modified the manuscript trying to respond to the various criticisms.

In detail:

R1: The authors should do an apoptotic assay to test that the concentrations of compound they are using are not causing apoptosis to any significant degree.

Reply: In agreement with the reviewer, we performed the apoptotic assay. Apoptosis was detected by staining A375 and MKN28 cells, after treatments, with Annexin-V-FITC/Propidium iodide (PI) solution followed by flow cytometry (FCM). The obtained results (Figure S3) demonstrated that the treatment with 10 µM of Ori or Iru did not induce any increment of apoptosis in both the cell lines, compared to the respective controls

R1: Scratch assays are known to cause cell damage in wound healing assays. I suggest the authors should perform a wound healing assay where an intermediate block is removed and the cells allowed to migrate into the space.

Reply: The wound healing assay is a standard in vitro technique to evaluate the cell migration in two dimensions. To stimulate wounding, the most common approach, widely used in literature even today [Li et al.2018; Pagliara et al. 2019], is to create a gap by scratching a confluent cell monolayer with a pipette tip. This approach is a good choice because it is inexpensive and easy to implement. I agree with the Reviewer that the physical exclusion methods (with plastic insert, for example) cause minimally damage to the remaining cells compared to the direct manipulation of the monolayer; however, it is also true that the damage is negligible and it slightly affect the assay result, since the same procedure is performed on both treated and control cells. Furthermore, to ensure the reproducibility of the wound, we scratched correctly tilting the pipette and applying constant pressure to create a constant gap width and repeating each single experiment at least in triplicate. Moreover, to photograph the same field during the image acquisition, we scraped horizontally and vertically with a pipette tip, to form a cross shaped wound. Cross shaped wounds were photographed on each well at 0, 6, 24 and 48 h by using an inverted microscope (10 × objective) and then quantitative analysis of the closure assay was performed by normalizing the gap area with the time 0 for each group and after normalization, the wound closure was expressed as percentage vs untreated control cells. This information were added in the revised version of the manuscript (see paragraph 4.9)

R1: Lines 124-125 – the authors say that Iru and Ori delay myotube formation. However, there is no evidence that this is delayed: this would require subsequent time point images that show myotube formation with a delayed effect. I suggest the authors provide such images of different time points, or otherwise change the conclusion to an inhibition of myoutube formation rather than a delay.

Reply: According to the referee suggestion, we changed “delayed” with “inhibited” (line 127)

R1: Lines 130-133 – the authors should say that Ori also had the same effect, and that the lesser effect with Ori (compared to Iru) in the Western blot is consistent with the myotube differentiation experiment.

Reply: In agreement with the Reviewer, we have inserted this comment into the text (lines 135-137)

R1. Line 206 – a brief explanation of DARTS in the main text would be useful for non-specialists

Reply: As suggested, we have included a description of the DARTS approach, explaining the principles on which it is based, and the techniques used to carry it out (lines 212-218.)

R1: Figures with blots – on all westerns, fold changes are written which is useful. But no errors/variation are included. It would be useful to know that such a change is significant across multiple experiments – this could be included as a supplementary bar graph with error bars, for example.

Reply: We have included a further supplementary figure (Figure S1), where the statistical analysis of the results obtained in triplicate experiments are reported.

R1: Discussion – the authors find a number of interesting candidates (table 1) but do not discuss them in their discussion section. A discussion of these would be very useful for the reader, as many of these candidates are known to be important in invasion (e.g. vinculin, tubulin, actin).

In accordance with the reviewer's suggestion, we have included a brief comment about the other possible targets of the two ent-kaurane diterpenes in the discussion (lines 436-443.)

Reviewer 2 Report

The manuscript provides an interesting characterisation of two compounds Oridonin and Irudonin. The authors perform cell-based experiments to investigate the impact of the compounds on cell viability. They further describe the impact on viability and cell motility in relation to cytoskeletal organisation.  They present data to suggest this is based upon targeting of Ezrin.

Specific points:

10-60 uM of the compounds are used. These are relatively high concentrations so I am concerned about multiple modes of action leading to an overall impact on cell viability – rather than specifically targeting a single protein. The authors should discuss this.

Moreover, in the methods is it claimed that purity is only >97%, at the high concentrations used in the assays the contaminants could be significant. Do the authors know what are the contaminants?

Figure 2. It is not clear how the fluorescence intensity was calculated – is this per area (ROI) or per cell. Staining with phalloidin can vary greatly cell to cell therefore it is possible to have changes in intensity. Interestingly, while intensity for Iru treated cells is lower, it appears that the cytoskeleton is still present?

Ori also has the biggest impact upon cell viability but it has less impact upon the cytoskeleton? Can the authors explain this in context of their conclusions.

Figure 3. The wound area appears to be very variable between conditions therefore how can the authors address wound area changes between conditions over time?

Figure 5. It would be useful to show quantification of the gels to calculate fold-change.

10 uM of each compound was used to examine the effect within A375 cell and MKN28 cells. Why was this concentration chosen? Especially when this concentration does not impact cell viability? I understand that it is not possible to perform assays at high concentrations but I am not convinced it is possible to see an impact at 10uM.

There are a few typos and minor edits required due to errors or lack of clarity.

This includes: Abstract line 32 – “disrupt” cytoskeleton organisation

Discussion line 391 “toto”

Scale bars – check to make sure all images have a scale bar and ensure that the bar is clearly visible.

Author Response

The authors thank the reviewer for his suggestions. we modified the manuscript trying to respond to the criticisms.

In detail:

R2: 10-60 uM of the compounds are used. These are relatively high concentrations so I am concerned about multiple modes of action leading to an overall impact on cell viability – rather than specifically targeting a single protein. The authors should discuss this.

Reply: The authors fully agree with this observation. In fact, the range of concentrations 10-60 µM was used exclusively to evaluate the general anti-proliferative activity of the two compounds. In conducting experiments aimed at studying the mechanism of action of the molecules, lower concentrations and / or shorter exposure times were chosen. Further clarification in this regard can be found below, in response to another reviewer's comment.

R2: Moreover, in the methods is it claimed that purity is only >97%, at the high concentrations used in the assays the contaminants could be significant. Do the authors know what are the contaminants?

Reply: The claim that the purity of the molecule is > 97% was used cautiously. Actually, the analyses of the two compounds carried out by HPLC-UV and NMR did not show us any other molecule. Taking into account the sensitivity limits of the two techniques, we did not consider it correct to state that the two compounds were 100% pure, but neither of the two compounds had contaminants detectable with the two techniques indicated above.

R2: Figure 2. It is not clear how the fluorescence intensity was calculated – is this per area (ROI) or per cell. Staining with phalloidin can vary greatly cell to cell therefore it is possible to have changes in intensity. Interestingly, while intensity for Iru treated cells is lower, it appears that the cytoskeleton is still present?

Reply: In agreement with Reviewer we have added the required information on fluorescence quantization in the Materials and Methods section (lines 567-571). Quantitative analysis was performed with imageJ, selecting an entire field as region of interest (ROI) because the cells were very confluent and differentiated, quantifying the mean pixel intensity per area. To reduce intrinsic variability, we repeated this measurement on at least thirty fields per experimental point. Then, the fluorescence intensity was relative to untreated cells, cultured in DM, set as 100%. The actin network is a dynamic structure with continuous directional polymerization and disassembly. The monomers of actin are regarded as globular-actin (G-actin), while the polymers are known as filamentous-actin (F-actin). Phalloidin, as also reported in data sheet, (Phalloidin, Tetramethylrhodamine B isothiocyanate, Catalog Number P1951, Merk) has been found to bind only to polymeric and oligomeric forms of actin, and not to monomeric actin. The treatment with Iru was able to reduce the polymeric actin formation (F-actin), and this leads to a lower fluorescence intensity after Phalloidin staining, but the cytoskeleton is present.

R2: Ori also has the biggest impact upon cell viability but it has less impact upon the cytoskeleton? Can the authors explain this in context of their conclusions.

Reply: We have inserted the following specific comments in the discussion (lines 422-428):

“Our results also allowed us to make an interesting comparison between the two investigated molecules. The two compounds differ only in the position of a hydroxyl group, thus it is not surprising that they are able to interact with the same target proteins. However, some differences between the two diterpenes emerged. Firstly, Iru has a higher affinity than Ori towards Ezrin and this determines its effectiveness in inhibiting the invasion capacity of tumor cells. On the other hand, Ori's toxicity could be associated with its high affinity for proteins, such as Hsp70 and Nucleolin, which play a key role in tumor cell survival.”

R2: Figure 3. The wound area appears to be very variable between conditions therefore how can the authors address wound area changes between conditions over time?

Reply: We thank the Reviewer for this observation, because it let me to explain this point better in the manuscript (lines 598-608).

“To ensure the reproducibility of the wound, we scratched with the pipette tip and applying constant pressure to create a constant gap width and repeating each single experiment at least in triplicate. Moreover, to photograph the same field during the image acquisition, we scraped horizontally and vertically with a pipette tip, to form a cross shaped wound. Cross shaped wounds were photographed on each well at 0, 6, 24 and 48 h by using an inverted microscope (10 × objective) and then quantitative analysis of the closure assay was performed by normalizing the gap area with the time 0 for each group. Then, after normalization, the wound closure was expressed as percentage vs untreated control cells.”

R2: Figure 5. It would be useful to show quantification of the gels to calculate fold-change.

Reply: We have included a further supplementary figure (Figure S1), where the statistical analysis of the results obtained in triplicate experiments are reported.

R2: 10 uM of each compound was used to examine the effect within A375 cell and MKN28 cells. Why was this concentration chosen? Especially when this concentration does not impact cell viability? I understand that it is not possible to perform assays at high concentrations but I am not convinced it is possible to see an impact at 10uM.

Reply: Experiments aimed at evaluating the effects of a treatment on the activity of specific proteins or on a specific pathway must be carried out using concentrations that do not induce anti-proliferative or pro-apoptotic effects. The preliminary tests we carried out (see Figure 3) had shown that already at a concentration of 20 µM both molecules induced anti-proliferative effects, which were instead absent when the treatment was carried out with a concentration of molecules equal to 10 µM. Moreover, previously reported result concerning the study of the effect of Oridonin on A375 cells were carried out using a compound concentration of 10 µM. Finally, CETSA results showed that EC50s of the two molecules in stabilizing Ezrin were between 13 and 21 µM (following 2h of treatment).Based of these considerations, we chose to use 10 µM of each compound.

However, in the new version of the manuscript we have included a brief explanation of the reasons that led us to choose this concentration (line 179-181)

R2: There are a few typos and minor edits required due to errors or lack of clarity.

This includes: Abstract line 32 – “disrupt” cytoskeleton organisation

Discussion line 391 “toto”

Reply: The text was further checked and corrected

R2: Scale bars – check to make sure all images have a scale bar and ensure that the bar is clearly visible.

Reply: We have included the scale bars in all the images, ensuring that they were visible.

Round 2

Reviewer 2 Report

I am happy for the manuscript to be accepted following the changes.